# TextTN: Probabilistic Encoding of Language on Tensor Network

## Abstract

As a novel model that bridges machine learning and quantum theory, tensor network (TN) has recently gained increasing attention and successful applications for processing natural images. However, for natural languages, it is unclear how to design a probabilistic encoding architecture to efficiently and accurately learn and classify texts based on TN. This paper proposes a general two-step scheme of text classification based on Tensor Network, which is named as TextTN. TextTN first encodes the word vectors in a probabilistic space by a generative TN (word-GTN), and then classifies a text sentence using a discriminative TN (sentence-DTN). Moreover, in sentence-DTN, its hyper-parameter (i.e., bond-dimension) can be analyzed and selected by the theoretical property of TextTN's expressive power. In experiments, our TextTN also obtains the state-of-the-art result on SST-5 sentiment classification task.

## 1 Introduction

Machine learning incorporating with the quantum mechanics forms a novel interdisciplinary field known as quantum machine learning (Huggins et al., 2019; Ran et al., 2020). Tensor network (TN) as a novel model has become prominent in the field of quantum machine learning (Biamonte et al., 2017). On the one hand, tensor network can be used as mathematical tool to enhance the theoretical understanding of existing neural network methods (Levine et al., 2018; 2019). On the other hand, based on tensor network, new machine learning algorithms have been proposed, e.g., discriminative TN (DTN) (Stoudenmire & Schwab, 2016) for supervised tasks and generative TN (GTN) (Han et al., 2018) for unsupervised scenarios (Han et al., 2018). Based on the natural analogy between the quantum concepts (e.g., quantum many-body system (Levine et al., 2018)) and the image representation, many studies and applications are conducted for processing and learning natural pictures (Stoudenmire & Schwab, 2016; Sun et al., 2020; Liu et al., 2019). However, for natural languages, it remains unclear how to design an efficient and effective TN approach, which can accurately learn and classify texts.

In the field of natural language processing (NLP), researchers have realized the analogy between the quantum many-body wave function and the word interactions (by the tensor product) in a text sentence, and developed a quantum-inspired language representation (Zhang et al., 2018). Based on the quantum many-body physics and tensor decomposition techniques, Zhang et al. (2018) provided a mathematical understanding of existing convolution neural network (CNN) based text classification methods. Similarly, a tensor space language model (TSLM) has been built based on the tensor network formulation (Zhang et al., 2019). This work shows that TSLM is a more generalized language model compared with $n-$gram and recurrent neural network (RNN) based language models. In implementation, however, TSLM did not provide a tensor network algorithm. The challenge lies in the high dimensionality of each word vector, which is much higher than the dimensionality of each pixel representation in image scenarios. After the tensor product of a number of word vectors, the resulting high-order tensors will become computationally intractable.

More recently, a tensor network algorithm, namely uniform matrix product state (u-MPS) model, has been proposed for probabilistic modeling of a text sequence (Miller et al., 2020). u-MPS is evaluated on a context-free language task, which uses an synthetic data set. However, u-MPS has not been applied in a real-world NLP task, e.g., typical language modeling or text classification task. In addition, the expressive power of u-MPS has not been investigated. The expressive power of tensor

network is a fundamental property of various TNs and has been systematically studied for tensor network factorizations of multivariate probability distributions (Glasser et al., 2019). This motivates us to make use of the theoretical property of TN's expressive power, for developing a tensor network based probabilistic model for natural language representation and classification [1].

To build such a text tensor network, we need to address two research problems in this paper. First, how to design a probabilistic encoding architecture to efficiently and effectively learn and classify the text. Second, how to analyse its expressive power, and make use of such analyses for more theoretical understanding and practical effectiveness of the text tensor network.

In this paper, we propose a novel tensor network architecture, named as TextTN. TextTN encodes each word vector in word-GTNs and classifies the sentence in a sentence-DTN. First, the proposed word-GTNs train a TN for each word and treat each element of a word vector as a node. In this manner, the word-GTNs firstly map a high-dimensional word vector a low-dimensional linear space by the tensor network operators. Then, the second layer tensor network, called as sentence-DTN, trains a TN for each sentence, by regarding the low-dimensional word vector obtained by word-GTNs as its input.

In TextTN, a sentence is represented by the tensor product among word vectors. Therefore, the interaction information among different word vectors and different dimensions are both modeled in TextTN. Such interactions, are encoded in the high-order weighted tensors, which represent a high-order semantic space. In both word-GTNs and sentence-DTN, the high-order tensor can be solved by the tensor network, i.e., a matrix product state model, which uses the idea of low-rank approximation that can conquer the exponential wall problem (Watson & Dunn, 2010).

In sentence-DTN, the bond-dimension is an important hyper-parameter and reflects the expressive power of TextTN. In this paper, we analyze the upper and lower bounds of the bond-dimension. Particularly, its lower bound can be determined by the entanglement entropy, which can be considered as a measurement of the communication information encoded in tensor network. A reference bond-dimension can be set as this lower bound, as we assume that a larger value means an information redundancy and a smaller value indicates an insufficiency of the TN's expressive power. In the experiments, such a reference bond-dimension can achieve effective classification results, which indicates the TextTN's practical advantage in its potential to save hyper-parameter tuning efforts.

Moreover, the word interaction has been taken into account in sentence-DTN by the joint effect of different words for the later class predication by the loss functions. For the learning algorithm, we observe that different word positions have different weights in a sentence, so that the one-function (for a specific position) training in the original DTN is inappropriate. Therefore, we propose an all-function training process in the sentence-DTN to improve the stability of TextTN.

We have evaluated TextTN in four major text classification datasets (MR, Subj, CR and MPQA). The results show that TextTN outperforms convolutional neural network (CNN) on all the datasets. This departs from vision tasks where according to the recent literature, a tensor network has not been reported to outperform CNN (Kim, 2014). In addition, based on the word vectors from the pre-trained model BERT, the TextTN has better results than the BERT model on SST-2 and SST-5 tasks, and the accuracy of BERT+TextTN is comparable with the state of the art (SOTA) result on SST-5 dataset.

## 2 BACKGROUND

We now provide the background of Matrix product States (MPS), a family of tensor networks. MPS (Schollwock, 2011) is also known as the tensor train decomposition (Oseledets, 2011). Because of the low degree of freedom, the research based on MPS is developing rapidly. At present, the tensor network based on MPS can be roughly divided into two categories. One is the Generative Tensor Network (GTN) (Han et al., 2018; Sun et al., 2020), and the other one is the supervised tensor network (also named as Discriminative Tensor Network, DTN) (Stoudenmire & Schwab, 2016). Then, we briefly describe existing GTN and DTN models for image classification tasks.

**GTNs** are used to model the joint probability distribution of given data. For a picture $X$ with $n$ pixels, each pixel is encoded into a two-dimensional vector $\boldsymbol{x}_i = (p_i, 1 - p_i)^T$ by a feature mapping from a

---

[1] In this paper, we focus on the text classification task. However, the idea and formulation of our proposed approach are general and have potential in other NLP tasks.

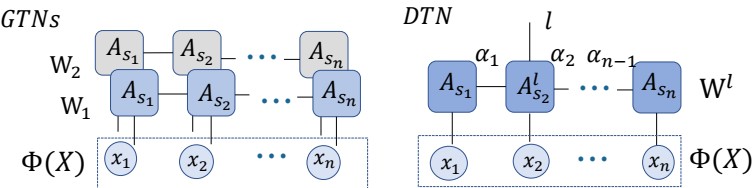

Figure 1: The schematic of tensor networks based on MPS. Left is GTNs and right is DTN.

pixel's value (Sun et al., 2020), where $i \in \{1, \ldots, n\}$ and $p_i$ is the mapped probabilities of the pixel $x_i$. The representation of the picture $X$ can be give out $\Phi(X)$ through the operator of tensor product between these vectors $\boldsymbol{x}_i$. A joint probability distribution of the picture $X$ is computed by the GTNs.

$$P_j(X) = |\mathbf{W}_j \bullet \Phi(X)|^2 \tag{1}$$

where $P_j(X)$ represents the probability of a picture $X$ with respect to the category $j$ ($j \in \{1, \ldots, m\}$), $m$ is the number of the categories on an image classification dataset, and $\bullet$ is the operator of tensor contraction. The MPS decomposition of the $j^{th}$ $n$-order weight tensor $\mathbf{W}_j$ can be written as:

$$\mathbf{W}_j = \sum_{\{\alpha\}} A_{s_1}^{\alpha_1} A_{s_2}^{\alpha_1 \alpha_2} \ldots A_{s_n}^{\alpha_{n-1}} \tag{2}$$

In the left of the Figure 1, we give out an illustrative GTNs in two categories (i.e., $m$=2). Second or third order tensors $A_{s_i}$ ($i \in \{1, \ldots, n\}$) are from the decomposition of the weight tensor $\mathbf{W}_j$. Each 'virtual' indicator $\alpha_k$ ($k \in \{1, \ldots, n-1\}$) is the rank obtained from the tensor-train decomposition (Oseledets, 2011), and the 'physical' indicator $s_i$ is the dimension of the pixel $x_i$.

**DTN** is used to identify the class or label of a picture and computes a conditional probability distribution $P(y|X)$. In DTN, the conditional probability distribution $P(y|X)$ is computed as follows.

$$P(y|X) = \mathbf{W}^l \bullet \Phi(X). \tag{3}$$

where the ($n$+1)-order weight tensor $\mathbf{W}^l$ is decomposed into a MPS and $l$ is an extra order/index representing the label:

$$\mathbf{W}^l = \sum_{\{\alpha\}} A_{s_1}^{\alpha_1} \ldots A_{s_i}^{l;\alpha_{i-1}\alpha_i} \ldots A_{s_n}^{\alpha_{n-1}} \tag{4}$$

where $P(y|X)$ is a vector and encodes the conditional probability distribution of outputting $y$ given a picture $X$. Eq. 3 is defined to classify the input $X$ by choosing the label for which the value in the vector $P(y|X)$ is largest. In practice, the MPS shown in the right of Figure 1 is a supervised learning model (Stoudenmire & Schwab, 2016), and these rank values $\{\alpha_k\}$ are set to be equal as the hyper-parameter (bond-dimension) in TN.

## 3 TENSOR NETWORK FOR LANGUAGE ENCODING AND CLASSIFICATION

**Problem setting** Our goal is to develop a tensor network architecture to encode and classify a sentence of words in a TN's probabilistic manner. For a sentence with $n$ words (i.e, $S$=($w_1, \ldots, w_n$)), the tensor representation of the sentence can be written as ($\boldsymbol{w}_1 \otimes \ldots \otimes \boldsymbol{w_i} \otimes \ldots \otimes \boldsymbol{w}_n$) (Zhang et al., 2019), where $\boldsymbol{w}_i$ are the word vectors. However, as aforementioned in Introduction, because of the dimensional disaster, it is infeasible to directly input such as a sentence representation into a tensor network. In order to solve this problem and still model the word interaction on a high-order tensor (denoted as $\mathbf{W}^l$), we can formalize the TN model as follows:

$$\begin{aligned} P(y|S) &= \mathbf{W}^l \bullet f(S) \\ &= \mathbf{W}^l \bullet (f(\boldsymbol{w}_1) \otimes f(\boldsymbol{w}_2) \otimes \ldots \otimes f(\boldsymbol{w}_n)) \end{aligned} \tag{5}$$

where $f$ is an operator to encode a sentence in a low-dimensional probabilistic space.

In our work, word-GTNs are used to encode the word vectors into a low-dimensional space, and then the new representation of word can be efficiently inputted to the tensor network. Specifically, the

function $f$ in Eq 5 is embodied as word-GTNs (in Section 3.1), and $\mathbf{W}^l$ is embodied as sentence-DTN (in Section 3.2). In the process of text classification, the word-GTNs and sentence-DTN are unified into a new TN framework (TextTN), as shown in Figure 2. Bond-dimension is an important hyper-parameter which reflects the expressive power of TN model and influences the effectiveness of text classification. In Section 3.2, we propose a reference bond-dimension that can be computed based on the entanglement entropy. Besides, to improve the stability of sentence-DTN, all-function learning on sentence-DTN is proposed in Section 3.3.

### 3.1 WORD ENCODING BASED ON WORD-GTNS

We unfasten TextTN step by step in the sentence representation and classification problem. First, $m$ word-GTNs are used to encode word vectors to a low-dimensional space, where $m(m > 1)$ corresponds to the dimension of the low-dimensional space. In Figure 2, as an example, two word-GTNs are used to encode each word in a sentence, with shared parameters for each word. Firstly, each element of word embedding vector is mapped by a feature mapping. Then, each word is represented as a $d$-order tensor by calculating the tensor product of each element after feature mapping. For example, for a word vector $\boldsymbol{w_i} = (\theta_1, \ldots, \theta_d)^T$, the tensor product representation is shown as follows:

$$\phi(\boldsymbol{w}_i) = \begin{pmatrix} \cos\left(\frac{\pi}{2}\theta_1\right) \\ \sin\left(\frac{\pi}{2}\theta_1\right) \end{pmatrix} \otimes \ldots \otimes \begin{pmatrix} \cos\left(\frac{\pi}{2}\theta_d\right) \\ \sin\left(\frac{\pi}{2}\theta_d\right) \end{pmatrix} \tag{6}$$

where $\otimes$ is the tensor product, and $\phi(\boldsymbol{w}_i)$ is the tensor product representation of word $w_i$. The feature mapping (i.e., $\cos(\cdot)$ and $\sin(\cdot)$) is motivated by "spin" vectors encoded in quantum systems, which is usually used in TN (Stoudenmire & Schwab, 2016; Han et al., 2018).

Then, two word-GTNs are used to encode the tensor $\Phi(\boldsymbol{w}_i)$ and get the probability distribution $\boldsymbol{v}_i$, $i \in \{1, \ldots, n\}$ (see Figure 2). Specifically, for word $w_i$, each word-GTN can encode the word into a probability value $P_j^i, j \in \{1, 2\}$. In the following, $\boldsymbol{v}_i$ is represented by a 2-dimensional vector $(P_1^i, P_2^i)^T$ which is calculated by:

$$\boldsymbol{v}_i = f(\boldsymbol{w}_i) = \begin{pmatrix} P_1^i \\ P_2^i \end{pmatrix} = \begin{pmatrix} |\mathbf{W}_1 \bullet \phi(\boldsymbol{w}_i)|^2 \\ |\mathbf{W}_2 \bullet \phi(\boldsymbol{w}_i)|^2 \end{pmatrix} \tag{7}$$

where $\bullet$ is the operator of tensor contraction, $\mathbf{W}_1$ and $\mathbf{W}_2$ are two weight tensors to be learned in word-GTNs, and shared for all word vectors in a sentence. The probability condition of $P_1^i + P_2^i = 1$ is then achieved by the $softmax$ in practice.

word-GTNs are different from the existing GTNs shown in Background section. The word-GTNs are used to learning the new word representation (rather than text representation), and the existing GTNs are used to classify the pictures (not for pixels). Since a word does not have a class or label, the output of word-GTNs corresponds to the probabilities in a low-dimensional feature space.

Regarding the dimension of low-dimensional space of word-GTNs. i.e, $m$, we will show that the upper bound of the bond dimension of TextTN is $m^{\lfloor n/2 \rfloor}$ in Theorem 1. The expressive power of the model will increase exponentially along with the increasing $m$ (Khrulkov et al., 2018). Therefore, the dimension $m$ should be set as a small value (e.g., 2 in this section) to prevent a too large expressive power. This observation is also supported by our experiments.

### 3.2 ANALYSIS OF BOND-DIMENSION ON SENTENCE-DTN

After we use word-GTNs, the new representation of a sentence $S$ can be written as $(\boldsymbol{v}_1 \otimes \ldots \otimes \boldsymbol{v}_n)$. which is a $n$-order tensor. We now introduce a sentence-DTN to classify sentences. This sentence-DTN is a MPS which is from the tensor-train decomposition (Oseledets, 2011) of high-order tensor. By inputting the sentence vector obtained by word-GTNs, the sentenc-DTN can be formalized as:

$$\begin{aligned} P(y|S) &= \mathbf{W}^l \bullet (\boldsymbol{v}_1 \otimes \ldots \otimes \boldsymbol{v}_n) \\ &= A_{S_1}(\boldsymbol{v}_1) A_{S_2}(\boldsymbol{v}_2) \ldots A_{S_i}^l(\boldsymbol{v}_i) \ldots A_{S_n}(\boldsymbol{v}_n) \end{aligned} \tag{8}$$

which is a $n+1$-order MPS. As we introduced in Background, we can set the rank values as an equal value (bond-dimension), i.e., $\alpha_1 = \ldots = \alpha_j = \ldots = \alpha_{n-1}$. In other words, the bond dimension is set as $r$

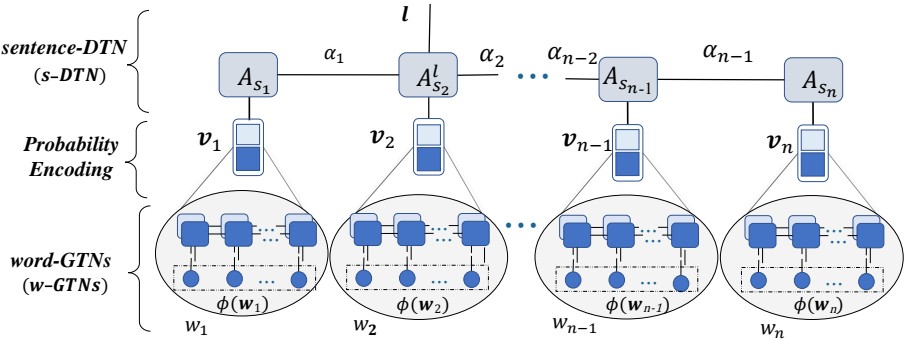

Figure 2: The diagram about the model architecture of TextTN.

$(r=\alpha_j)$. Then, each element of $\mathbf{W}^l$ can be calculated as follows,

$$\mathbf{W}^l{}_{S_1,\dots,S_n} = \sum_{\{\alpha_j=1\}}^{r} A_{s_1}^{\alpha_1}\dots A_{s_i}^{l;\alpha_{j-1}\alpha_j}\dots A_{s_n}^{\alpha_{n-1}} \tag{9}$$

The bond-dimension $r$ is the hyper-parameter which can reflect the expressive power of a tensor network. We provide an approach to a reference bond-dimension, which is an optimal parameter setting reported in our experiments. In the following, we will derive the bound of the bond-dimension and show that its lower bound is the *reference* or *recommended* bond-dimension. This lower-bound, is calculated by the entanglement entropy, which will be described as follows.

**Definition 1.** *For an $n+1$-order weight tensor $\mathbf{W}^l$ decomposed to a matrix product states (MPS), $\lambda$ ($\lambda \leq r$) singular values can be given out in the $\lfloor \frac{n}{2} \rfloor$-th node of MPS. The entanglement entropy is calculated by the singular-value vector $\mathbf{g}_k$ as*

$$E = -\sum_{k=1}^{\lambda} \mathbf{g}_k^2 \log \mathbf{g}_k{}^2 \tag{10}$$

*where $\lambda$ is the number of non-zero singular values, log is the log to the base 2, and the normalization condition of the singular-value vector $\sum_{k=1}^{R} g_k^2 = 1$ is satisfied.*

Entanglement entropy measures the amount of communication information in TN (Levine et al., 2019), and we can derive the lower bound of the bond-dimension based on entanglement entropy, as shown in Theorem 1.

**Theorem 1.** *For the input $\{\mathbf{v}_1,\dots,\mathbf{v}_n\}$ of the sentence-DTN, the bond-dimension is bounded as*

$$m^{\lfloor n/2 \rfloor} \geq bond-dimension \geq \lfloor 2^E \rfloor. \tag{11}$$

*where $E$ is the entanglement entropy of TextTN, $m$ is the inputting dimension of sentence-DTN, $n$ is the length of a sentence, and $\lfloor\ \rfloor$ indicates rounding down.*

*Proof.* The proof can be found in Appendix B. □

According to Eq. 11, the lower bound of bond-dimension can be determined by the entanglement entropy $E$. If the bond-dimension is less than $\lfloor 2^E \rfloor$, TextTN can not model sufficient information measured by the entanglement entropy $E$. Compared with the lower bound $\lfloor 2^E \rfloor$, choosing a larger value which satisfies Eq.11 will increase the number of parameters and the whole computational cost of the TextTN, but only gain little information from the data. In addition, the increase of the bond-dimension will increase the model complexity, possibly leading to the overfitting problem. Therefore, we recommend the lower bound $\lfloor 2^E \rfloor$ as the reference bond-dimension.

About calculating the required entanglement entropy $E$, we first need to use a sufficiently large bond-dimension. Since the initial bond-dimension is large enough and the entanglement entropy finally converges, the $E$ measures the amount of information that the network can capture from a

certain data set (Levine et al., 2019). After calculating the entanglement entropy $E$, according to the proof of the Theorem 1, we get the lower bound of bond-dimension shown as Eq.11, which is the reference bond-dimension.

Moreover, we can derive the upper bound of the bond-dimension. According to the research (Khrulkov et al., 2018), the upper bound of the bond-dimension can be given out and is $m^{\lfloor n/2 \rfloor}$, where $m$ is the inputting dimension of each node in MPS, and $n$ is the number of the nodes, i.e., the length of sentence in TextTN. If the bond-dimension is greater than $m^{\lfloor n/2 \rfloor}$, then the network structure will not meet the constraints of Tensor-Train decomposition (Oseledets, 2011) ( Appendix B), then TextTN will be rank deficient.

### 3.3    ALL-FUNCTION LEARNING ON SENTENCE-DTN

In the original DTN, the position of $l$ in Figure 2 can be moved from $A_{S_1}$ to $A_{S_n}$ in the process of training. It means that any position can be selected for the label bound $l$, leading to a variance for different positions in the processes of training and prediction. The following Claim 1 shows that the original DTN calculates a conditional probability by putting the label bound in a certain of the model, which corresponds to a certain certain probability function. However, the effects of putting the label bound in different positions should be taken into consideration.

**Claim 1.** *For an input $S = \{\boldsymbol{v}_1, \boldsymbol{v}_2, \ldots, \boldsymbol{v}_n\}$, the original DTN (Stoudenmire & Schwab, 2016) computes the conditional probability $P(y|S)$ by putting the label bound $l$ in a certain position, which can be considered as a function from a set $T$ of functions which compute all the possible conditional probabilities. Such a set is defined as:*

$$T = \{P(y|\boldsymbol{v}_1; \boldsymbol{v}_2 \ldots \boldsymbol{v}_n), \ldots, P(y|\boldsymbol{v}_1 \ldots \boldsymbol{v}_i; \boldsymbol{v}_{i+1} \ldots, \boldsymbol{v}_n), \ldots, P(y|\boldsymbol{v}_1 \ldots \boldsymbol{v}_{n-1}; \boldsymbol{v}_n)\}, \quad (12)$$

*where $i$ the location of segmentation for the input S, and is also a position of label $l$.*

*Proof.* The proof can be found in Appendix C. □

In the original DTN (Stoudenmire & Schwab, 2016), the loss function is defined as $L(\mathbf{W}^l) = \sum_{j=1}^{N} l(\langle f(S), \mathbf{W}^l \rangle, \boldsymbol{y})$ based on the label bound $l$ in a certain position, which only corresponds to a certain probability function. If one function is learned, the different aspects probability distribution for the label bound $l$ in different position can not be considered. To solve this problem, we propose to model all functions in the set $T$ in Eq. 12, with a new loss defined as follows:

$$L(\mathbf{W}^l, \mathbf{g}) = \frac{1}{N} \sum_{j=1}^{N} CE(\sum_{i=1}^{n} \mathbf{g_i} \langle f(S), \mathbf{W}^{l_i} \rangle, \boldsymbol{y}) \quad (13)$$

where $CE$ is cross-entropy, $\boldsymbol{y}$ is the label that is represented by a one-hot vector, $N$ is the number of samples, $n$ is the length of the sentence, $\mathbf{g}$ is a parameter vector and $\mathbf{g_i}$ is a value in the vector. $\langle f(S), \mathbf{W}^{l_i} \rangle$ is one function from the function set $T$. The sentence-DTN with all-function learning is expected to achieve higher accuracy than the original DTN.

Finally, we give a Algorithm to describe our training method of overall TextTN given a sentence $S$ with $n$ words in the **Algorithm 1**.

## 4    EXPERIMENTS

In this section, we divide the experiment into two parts based on the theories presented above. The first set of experiments is to verify the effectiveness of TextTN. The second set of experiment is to verify that the effectiveness of the reference bond-dimension, which can be computed by Eq. 11 based on the entanglement entropy for different datasets. A Python implementation of the proposed algorithm is placed in Supplementary Material.

### 4.1    TEXT CLASSIFICATION EVALUATION OF TEXTTN

**Datasets and Benchmarks**: Six text classification datasets are used in our experiments. **MR** (Pang & Lee, 2004b): Movie reviews are divided into positive and negative categories; **CR** (Hu & Liu,

---

**Algorithm 1** Training Algorithm of TextTN

---

**Input:** $S = \{w_1, \ldots, w_n\}, w_i \in \mathbb{R}^d$ (word embedding vectors), $y \in \mathbb{R}^l$ (the label of $S$);
**Output:** $\mathbf{W}_1, \mathbf{W}_2$ and $\mathbf{W}^l$ (parameters tensors);
  1: Initialize $\mathbf{W}_1$ and $\mathbf{W}_2$ ( for word-GTNs); $\mathbf{W}^l$ (for sentence-DTNs);
  2: Initialized training : the reference bond-dimension for $\mathbf{W}^l$ is obtained by the process.
  3: **repeat**
  4:     Feature mapping: $w_i \in \mathbb{R}^d \longrightarrow \phi(w_i) \in \mathbb{R}^{d \times 2}$;
  5:     Generate probability codings: $v_i = softmax(|\mathbf{W}_1 \bullet \phi(\boldsymbol{w}_i)|^2, |\mathbf{W}_2 \bullet \phi(\boldsymbol{w}_i)|^2), v_i \in \mathbb{R}^2$;
  6:     Compute conditional probability; $\overline{y} = \mathbf{W}^l \bullet (v_1, \ldots, v_n), \overline{y} \in \mathbb{R}^l$;
  7:     Use Cross-Entropy method (CE) to calculate loss: $\min L = CE(\overline{y}, y)$;
  8:     Perform backpropagation by $L$ and update parameters $\mathbf{W}_1, \mathbf{W}_2$ and $\mathbf{W}^l$;
  9: **until** The loss $L$ converges

---

2004): Customer reviews set where the task is to predict positive or negative product reviews; **Subj**: Subjectivity dataset where the target is to classify a text as being subjective or objective; **MPQA** (Wiebe et al., 2005): Opinion polarity detection subtask; **SST-5** (Pang & Lee, 2004a): The movie reviews in the Stanford Sentiment Treebank, which is a fine-grained classification task (negative, somewhat negative, neutral, somewhat positive, positive); **SST-2** (Socher et al., 2013): Stanford Sentiment Treebank with binary classes. The evaluation metric is "Accuracy" of all tasks.

For CR, MR, MPQA, and Subj tasks, we use publicly word vectors (Word2vec) (Pennington et al., 2014) with 300 dimensions. For SST-2 and SST-5, the Bert-Large pre-trained model (Devlin et al., 2019) is used to obtain word vectors, and the dimension of the word vector from the BERT is 1024. More dataset statistics and experimental settings are given out in the Appendix D.1.

Table 1: The comparison of experimental results between TextTN and other benchmarks. Accuracy is the evaluation metric.

| Model | MR | CR | Subj | MPQA |
|---|---|---|---|---|
| CNN | 81.5 | 85.0 | 93.4 | 89.6 |
| DiSAN | – | 84.8 | 94.2 | 90.1 |
| Capsule-B | **82.3** | 85.1 | 93.8 | – |
| SGC | 75.9 | – | – | – |
| MPSAN | – | 85.4 | 94.6 | **90.4** |
| TextTN | 82.2 | **85.7** | **95.3** | **90.4** |

Table 2: The comparison between TextTN and other complicated models based on pre-trained model.

| Model | SST-5 | SST-2 |
|---|---|---|
| BCN+ELMo | 54.7 | – |
| Star-Transformer | 53 | – |
| FLOATER-large | – | **96.7** |
| CNN_Large | – | 94.6 |
| BERT | 52.9 | 94.6 |
| BERT+TextTN | **54.8** | 95.3 |

Table 3: Ablation experiments. The comparative results between TextTN without all-function learning (TextTN w/o all-function), TextTN without word-GTNs (TextTN w/o w-GTNs), and TextTN on four classification datasets (CR, Subj, MPQA and MR). The accuracy is adopted as the evaluation metric.

| Model | MR | CR | Subj | MPQA |
|---|---|---|---|---|
| TextTN w/o w-GTNs | 78.6 | 80.2 | 92.2 | 88.0 |
| TextTN w/o all-function | 80.5 | 84.1 | 94.4 | 89.8 |
| TextTN | 82.2 | 85.7 | 95.3 | 90.4 |

Ten benchmarks are used to compare with TextTN. In Table 1, **CNN** (Kim, 2014) is used for classification; **DiSAN** (Shen et al., 2017) is a self-attention based model; **Capsule-B** (Yang et al., 2018) is a capsule neural network; **SGC** (Wu et al., 2019) proposes a graph neural network; **MPSAN** (Dai et al., 2020) designs a novel self-attention network. In Table 2 , **BCN+ELMo** (Peters et al., 2018) is a a deep bidirectional language model; **Star-Transformer** (Guo et al., 2019) is a improved Transformer network; **BERT** (Devlin et al., 2019) , **CNN_Large** (Baevski et al., 2019) and **FLOATER** (Liu et al., 2020) are complicated neural network models based on the BERT_Large pre-trained model.

**Experimental Results and Analysis**: First, TextTN is experimented on four datasets. The results in Table 1 show that TextTN has better results than CNN (Kim, 2014) on MR (+0.7%), CR (+0.7%),

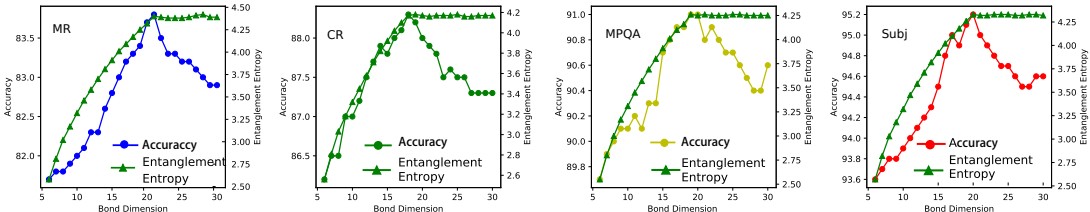

Figure 3: On four classification tasks (MR, CR, MPQA and Subj), we calculate the entanglement entropy and evaluate the accuracy of TextTN, respectively, with different bond-dimensions set from 5 to 30.

Subj (+1.9%) and MPQA (+0.8%). Moreover, compared with the state-of-the-art self-attention based model MPSAN (Dai et al., 2020), TextTN outperforms its results on CR by 0.3%, Subj by 0.7%, and obtains a same result 90.4% on MPQA.

After that, TextTN is experimented with the pre-training word vectors from BERT on SST-2 and SST-5. The results are shown in Table 2. In this experiment, BERT+TextTN achieves the comparable result with the model ELECTRA (Clark et al., 2020), and obtains higher results than BERT (Devlin et al., 2019) by 0.7% on SST-2. For SST-5 dataset, BERT+TextTN outperforms the state-of-the-art results 54.7 by 0.1%, and is higher than BERT (Devlin et al., 2019) by 1.9%.

In order to verify the effectiveness of all-function learning and word-GTNs, we perform ablation experiments shown as Table 3. First, when TextTN is without the all-function learning (TextTN w/o all-function), the classification accuracy declines to varying degrees on the four data sets. The accuracy values on MR, CR, Subj and MPQA drop by 1.7%, 1.6%, 0.9% and 0.6%, respectively. The results verify that the method of all-function learning for text classification is effective and useful.

In addition, in Table 3, we evaluate the effectiveness of word-GTNs by ablation experiments. TextTN w/o w-GTNs means that we use a linear function of the word embeddings to obtain the low-dimensional word representations, which are inptted to the sentence-DTN. On the contrary, TextTN uses word-GTNs to generate sentence representations. Compared with TextTN, the accuracy of TextTN w/o w-GTNs are largely reduced on four dataset. In particular, CR drops by 5.5%, MR, MPQA, and Subj decrease by 3.6%, 3.1%, and 2.4%, respectively. This shows that word-GTNs can effectively model word dependence in sentences and provide more effective semantic information for the classifier.

Finally, we analyze the impact of different word vector dimensions on the performance of TextTN, and the results are shown in the Appendix D.2. In addition, in the Section 3.1, the dimension $m$ of probability encoding is a hyper-parameter set to 2. In order to verify this choice, we conduct a comparative experiment, and the results in Appendix D.3 illustrate that compared with $m = 2$, the accuracy of classification with $m > 2$ has not been improved, or even decreased.

## 4.2 EVALUATION OF THE REFERENCE BOND-DIMENSION OF TEXTTN

In Theorem 1, we show that the lower-bound of the bond-dimension can be derived by the entanglement entropy, and then consider such a lower bound can be the reference bond-dimension of TextTN. In order to evaluate such a reference hyper-parameter value, we designed experiments as follows. First, we can compute the entanglement entropy through the initialized training of TextTN on text classification datasets (see Appendix D.4 for details). After that, based on the entanglement entropy, the reference bond-dimension can be computed.

For MR, CR, MPQA and Subj datasets, their accuracy scores are highest when the bond-dimensions of sentence-DTN are 21,18,19 and 20, respectively. These values (i.e., 21,18,19 and 20), are actually the reference bond-dimensions, which are computed based on the entanglement entropy from the initialized training. Specifically, these values can be computed by Eq. 11, i.e., $\lfloor 2^{4.40} \rfloor = 21$, $\lfloor 2^{4.20} \rfloor = 18$, $\lfloor 2^{4.32} \rfloor = 19$ and $\lfloor 2^{4.37} \rfloor = 20$, respectively. In Figure 3, four charts show that the accuracy values reach the highest point when the reference bond-dimensions are computed as 21(MR), 18(CR), 19(MPQA) and 20(Subj).

As shown in Figure 3, by reducing and increasing the bond-dimension with respect to its reference value, we also observe interesting results about the accuracy. For example, in the MR task, when bond dimension $\leq 21$ accuracy is less than the accuracy when the bond-dimension is the reference value 21. If bond dimension $\geq 21$, the accuracy begins to decline, but the updated entanglement entropy value (computed by the non-zero single values of sentence-DTN) remains stably. The same results are shown on the other three data sets, i.e., CR, MPQA and Subj.

In summary, the above results show the effectiveness of the reference bond-dimension. However, a more rigorous analysis about the reference bond-dimension and the classification effectiveness of TextTN is still an open research question.

## 5    Conclusion and Future Work

In this paper, we propose a text tensor network, named as TextTN, which aims at encoding each word (in a sentence) from a high dimensional vector to a low-dimensional vector by word-GTNs. After that, the new sentence representations can be inputted to the sentence-DTN to carry out the sentence classification. The proposed TextTN achieves better results than CNN, and even better experimental results than some neural network models, such as HCapsNet. In addition, TextTN also achieves the experimental results that largely exceeds GTNs in all datasets. The entanglement entropy is defined in TextTN and is helpful to identify the the reference bond-dimension which usually achieves the best accuracy in our experiments.

In the future, we would like to further investigate more theoretical evidence and understanding of using of the entanglement entropy to set an optimal bound dimension. In the practical point of view, we can apply TextTN on more NLP tasks and test the effectiveness of both TextTN and its reference hyper-parameters.

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

## A  BASIC KNOWLEDGE ABOUT TENSOR NETWORK (TN)

We now describe some basic knowledge on tensor and tensor contraction, as well as the matrix product states (MPS).

### A.1  TENSOR AND TENSOR CONTRACTION

A tensor can simply be thought of as multi-dimensional array. The order of a tensor is defined to the number of indexing entries in the array. For example, in Figure 4a, the vector $v$ with the index $i \in m^2$, is a one order tensor, and the matrix $\mathbf{A}$ with two indexes (or nodes) $i \in m$ and $j \in n$ is

---

[2]$i \in [m]$ also is written as $i \in \{1, 2, \ldots, m\}$

a two-order tensor. After that, a n-order tensor is drawn in Figure with n nodes, $d_1, d_2, ...,$and $d_n$, where each node corresponds to an input on TN.

Tensors have some basic operations. The most fundamental one is tensor contradiction, which means summing over same indices of two tensors. Two examples are presented here in order to illustrate the operation more directly. In Figure 4a, we show the process of a matrix (2-order tensor) contracting with a vector (1-order tensor). In Figure 4b, another contraction process between two matrix **A** and **B** is demonstrated. And those high-order tensors, the tensor contraction operations between them follow the same rules with the above examples.

### A.2    MATRIX PRODUCT STATE (MPS)

A Tensor Network as a weighted graph, where each node corresponds to a tensor. It has a linear architecture with a list of nodes and edges, and the order of the tensor is the degree of the node in the graph. The weight of each edge is referred to as its bond dimension. In (Novikov et al., 2016; Han et al., 2018)), MPS is a kind of TN, which is decomposed by a high-order tensor through Tensor-Train decomposition (Oseledets, 2011). First, as shown in Figure 5a, the high-order tensor $W$ is decomposed into multiple low-order tensors. Then, through tensor contraction, the low-order tensors represented by nodes can be contracted into a MPS network in Figure 5b. The formula is as follows:

$$\mathbf{W} = \sum_{\{\alpha\}} A_{d_1}^{\alpha_1} \dots A_{d_i}^{\alpha_{i-1}\alpha_i} \dots A_{d_N}^{\alpha_{N-1}} \tag{14}$$

In the MPS, $\alpha_1, \alpha_2, \dots, \alpha_{N-1}$ are the indexes of bond_dimensions (also named as TT-rank in some works), $d_i \in \mathbb{R}^m$ $(i \in [N])$. If the low-order tensors is not trimmed in the process of TT decomposition, that is, the full rank is guaranteed, then $\alpha_1 \in \mathbb{R}^m, \alpha_2 \in \mathbb{R}^{m^2}, \alpha_{\lfloor N/2 \rfloor} \in \mathbb{R}^{m^{\lfloor N/2 \rfloor}}$, $\dots, \alpha_N \in \mathbb{R}^m$. That is, the maximum bond-dimension is $m^{\lfloor N/2 \rfloor}$.

Therefore, we avoid directly using the high-order tensor and Tensor-Train decomposition due to its exponentially increasing parameters. During the practical process, we do not construct a high-order tensor, nor do we perform tensor decomposition. Instead, we usually initialize an MPS architecture with random generated low-order tensors, optimizing them while training in order to better approximate feature distribution of data (which can be understood as a high-order tensor). Within the allowable deviation range, the amount of parameters will be dramatically reduced from $m^N$ to $N \times m \times r \times r$, and $r$ is the value of bond-dimension.

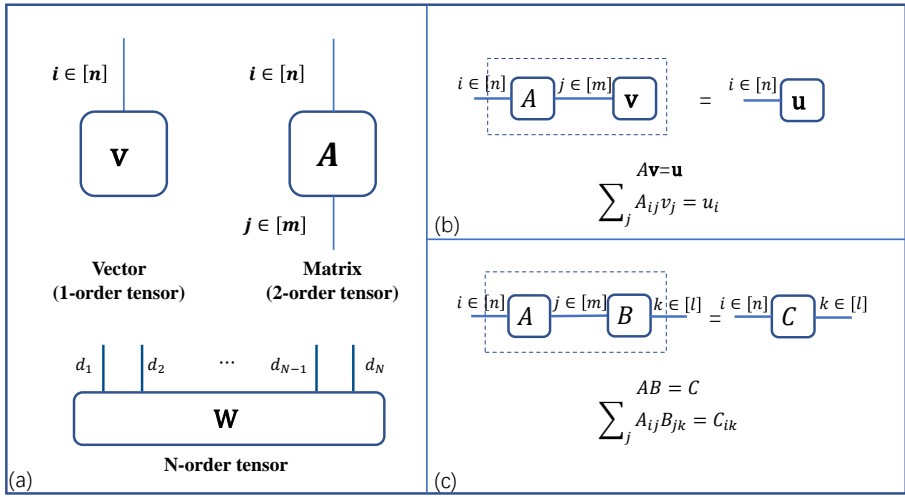

Figure 4: The schematic diagram of tensors and tensor contraction

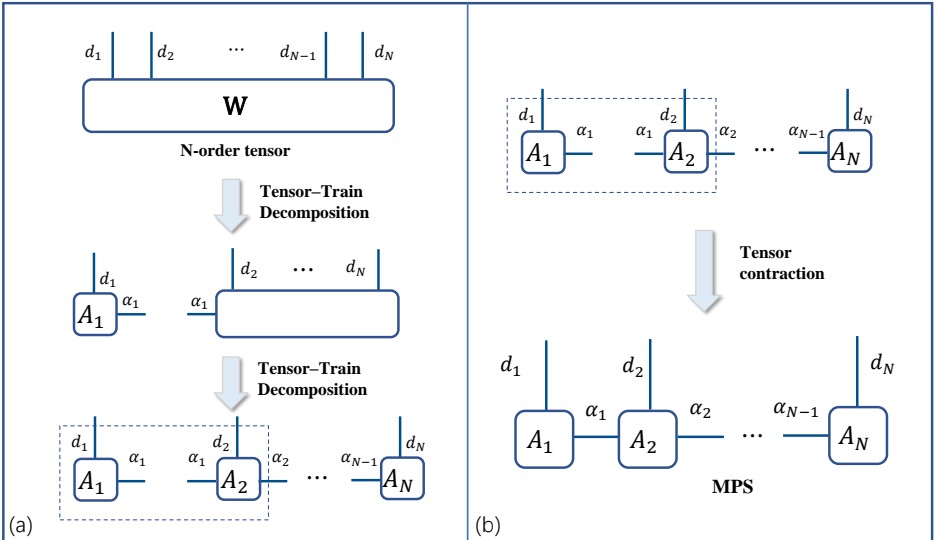

Figure 5: The schematic diagram of Tensor-Train decomposition and Matrix Product State

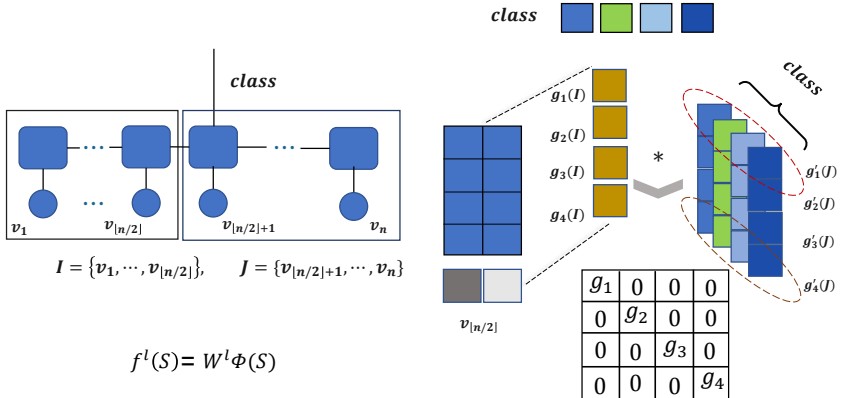

Figure 6: The diagram about the computing of singular values for the entanglement entropy. The left is the position of separation in sentence-DTN (the position is $v_{\lfloor n/2 \rfloor}$), and the right is the representation of singular values in the process of SVD for sentence-DTN.

## B  PROOF OF THEOREM 1

In TextTN, the bond-dimension is an important hyper-parameter which directly influences the expressive power of tensor network models (i.e., TextTN). An reference bond-dimension can be given out based on the entanglement entropy which is defined in **Definition 1.**, Therefore, The Theorem 1 shows the relation between the entanglement entropy and the bond-dimension, and gives out the lower-bond and the upper-bond of the bond-dimension. The proof of inequality 15 is showed here.

**Theorem 1.** For the input of the sentence-DTN $\{v_1, \ldots, v_n\}$, the bond- dimension is bounded as

$$m^{\lfloor n/2 \rfloor} \geq bond-dimension \geq \lfloor 2^E \rfloor. \tag{15}$$

where $E$ is the entanglement entropy of TextTN, $m$ is the inputting dimension of sentence-DTN, $n$ is the length of a sentence, and $\lfloor \ \rfloor$ indicates rounding down.

*Proof.* As for the upper bound of the Eq. 15, as shown in Appendix A.2, sentence-DTN as a MPS tensor network can be represented by a decomposed n-order tensor. If no rank is cut during

decomposition, the bond-dimension (TT-rank) of sentence-DTN is the same as the $n$-order tensor. Suppose the formula of n-order tensor is:

$$\mathbf{T} = (v_1 \otimes \ldots \otimes v_{\lfloor n/2 \rfloor}) \otimes (v_{\lfloor n/2 \rfloor+1} \otimes \ldots \otimes v_n) \tag{16}$$

$v_i$ is the word probability encodings shown in Eq.7. $\mathbf{T}$ is a $m^n$ tensor. From the matrix maximum rank theorem, the maximum rank in $\mathbf{T}$ appears at the position $\lfloor n/2 \rfloor$, that is, the maximum rank is $m^{\lfloor n/2 \rfloor}$. Thus, the upper bound of bond-dimension in sentence-DTN is also $m^{\lfloor n/2 \rfloor}$.

As for the lower bound of Eq. 15, the entanglement entropy $E$ is first computed by the first initialization training. For the lower bond of bond-dimension, practically, the singular values between $\boldsymbol{v}_{\lfloor n/2 \rfloor}$ and $\boldsymbol{v}_{\lfloor n/2 \rfloor+1}$ in Figure 6 can be obtained in the training process of TextTN, when TextTN converges.

Assume that $bond-dimension$ of TextTN is equal to $k$. The maximal entanglement entropy is computed when the $k$ singular values in TextTN are equal. Eq. 17 gives out the maximal entanglement entropy based on a fixed bond-dimension $k$.

$$E = -\sum_{i=1}^{k} \frac{1}{k} \log \frac{1}{k} = \log k, \tag{17}$$

On both sides of the equation, doing the exponential calculations in base 2 on both sides of the equation, the new equation can be computed as follows,

$$2^E = k. \tag{18}$$

When $k$ singular values are not equal, the inequality can be written as

$$k \geq 2^E. \tag{19}$$

To ensure that $bond-dimension$ represented by $k$, is an integer, the inequality can rewritten as $k \geq \lfloor 2^E \rfloor$, where $\lfloor \; \rfloor$ indicates rounding down.

Therefore, the inequality $bond-dimension \geq \lfloor 2^E \rfloor$ in Eq 15 can be established. Recall that the entanglement entropy $E$ computed by Eq. 17 represents the maximal amount of information that TextTN can model. Eq. 19 implies that the maximal entanglement entropy $E$ determines the lower bound of the bond-dimension of TextTN. For the upper bond, a parameter tensor $\mathbf{W} \in \mathbb{R}^{\mathbf{m^n}}$ can be viewed as a matrix with the shape $m^i \times m^{n-i}$. The maximal rank can be obtained when the parameter tensor $\mathbf{W}$ is viewed as the matrix with the shape $m^{\lfloor n/2 \rfloor} \times m^{n-\lfloor n/2 \rfloor}$. In this case, the value of maximal rank can be equal to $m^{\lfloor n/2 \rfloor}$, which is the upper bond of bond-dimension.

Therefore, the upper bond and lower bond of bond-dimension be obtained.

$\square$

Theorem 1 has also been validated in practice. In the process of TextTN training, we can compute the entanglement entropy of TextTN according to the Definition 1. When this training process converges, the maximal entanglement entropy is obtained. In addition, TextTNs with different bond-dimensions are experimented. At the beginning, the classification accuracy of TextTN increases with the increasing of the bond-dimension. When the bond-dimension increases to a certain value, the accuracy of TextTN begins to decline slowly. Such a certain value is the reference bond-dimension and can be computed by $\lfloor 2^E \rfloor$, when the training process converges.

## C    PROOF OF CLAIM 1

**Claim 1** For an input $S = \{\boldsymbol{v}_1, \boldsymbol{v}_2, \ldots, \boldsymbol{v}_n\}$, the original DTN (Stoudenmire & Schwab, 2016) computes the conditional probability $P(y|S)$ by putting the label bound $l$ in a certain position, which can be considered as a certain function from a set $T$ of functions which compute all the possible conditional probabilities. Such a set is defined as:

$$T = \{P(y|\boldsymbol{v}_1; \boldsymbol{v}_2 \ldots \boldsymbol{v}_n), \ldots, P(y|\boldsymbol{v}_1 \ldots \boldsymbol{v}_i; \boldsymbol{v}_{i+1} \ldots, \boldsymbol{v}_n), \ldots, P(y|\boldsymbol{v}_1 \ldots \boldsymbol{v}_{n-1}; \boldsymbol{v}_n)\}, \tag{20}$$

where $i$ the location of segmentation for the input $S$, and is also a position of label bound $l$.

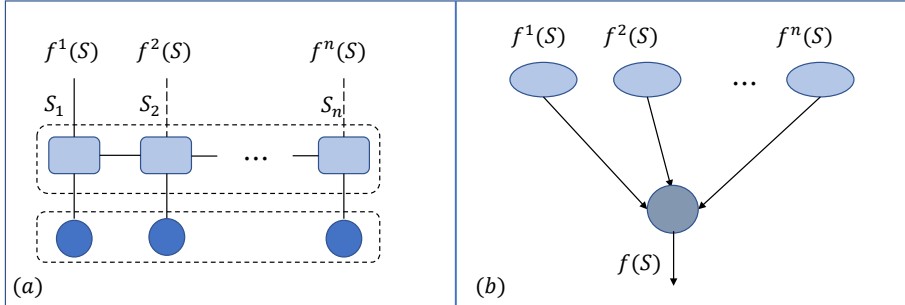

Figure 7: The diagram of all-function learning for tensor network. $(a)$ The conditional probability functions for the output in different position $S_i (i \in \{1, \ldots, n\})$. $(b)$ Getting an all-function $f(S)$ by combining all functions $f^i(S)$ from different positions.

*Proof.* For an input $S=\{\boldsymbol{v}_1, \ldots, \boldsymbol{v}_n\}$, which is from the output of word-DTNs, a $n$-order tensor, i.e., the input of the sentence-DTN, is given b y the operator of tensor product between vectors, $\boldsymbol{v}_i$, $i \in \{1, \ldots, n\}$. The parameter tensor $\mathbf{W}^l$ in DTN is a $(n + 1)$-order tensor. There is a mode (or an index) in tensor $\mathbf{W}^l$ to control the number of class in the classification task, which is $l$. As shown in Figure 7 $(a)$, in the processing of training, the index $l$ is moved from $S_1$ to $S_n$.

When the position of $l$ is $S_1$, the input $S$ is split two parts, which are $I=\{\boldsymbol{v}_1\}$ and $J=\{\boldsymbol{v}_2, \ldots, \boldsymbol{v}_n\}$, the basic events are $I$ and $J$, the classification function $f^1(S)$ can be represented by condition probability as follows,

$$f^1(S) = P(y|I, J) = P(y|\boldsymbol{v}_1; \boldsymbol{v}_2 \ldots \boldsymbol{v}_n). \tag{21}$$

When the position of separation $i$ is $1<i<n$, the input $S$ is separated into two parts, which are $I=\{\boldsymbol{v}_1, \ldots, \boldsymbol{v}_i\}$ and $J=\{\boldsymbol{v}_{i+1}, \ldots, \boldsymbol{v}_n\}$, the function $f^1(S)$ is represented

$$f^i(S) = P(y|\boldsymbol{v}_1 \ldots \boldsymbol{v}_i; \boldsymbol{v}_{i+1} \ldots \boldsymbol{v}_n). \tag{22}$$

When the position of separation $i$ is $i=n-1$, the function $f^1(S)$ is denoted Eq. 23 similar to Eq. 21 and Eq. 22.

$$f^{n-1}(S) = P(y|\boldsymbol{v}_1 \ldots \boldsymbol{v}_{n-1}; \boldsymbol{v}_n). \tag{23}$$

For different functions $f^i(S)$, they represent different condition probabilities. In the existing work (Stoudenmire & Schwab, 2016), i.e., original DTN, one function is used to classify the input $S$. □

In order to model all functions in the process of training, and one function learning cannot model all conditional probabilities in the input $S$, we propose all-function learning method for text classification, which is showed in Figure 7 $(b)$. We adopt weighted summation for these functions (i.e., $f(S)=\sum_{i=1}^{n} \mathbf{g_i} f^i(S)$, where $\mathbf{g}$ is a parameter vector, $\mathbf{g_i}$ is an element from the vector $\mathbf{g}$).

# D EXPERIMENTAL RESULTS AND ANALYSIS

In this section, we first describe the datasets used in our experiments.

## D.1 DATASETS

The statistics of six datasets are showed in Table 4. For CR, MR, MPQA and Subj datasets, they all have a total file. Therefore, we randomly select $10\%$ of the file as the validation set and test set, respectively, and select $80\%$ of the file as the train set. The evaluating indicator of all text classification tasks is the "Accuracy".

For CR, MR, MPQA and Subj tasks, we implement our model with Pytorch-1.20, and train them on a Nvidia P40 GPU. As for learning method, we use the Adam optimizer Kingma & Ba (2015) and an exponentially decaying learning rate with a linear warm up. For SST-2 and SST-5 dataset, the initialized learning rate is 5e-5.

Table 4: Statistics of all datasets. $Metrics$: Evaluation Metric, $L$: maximum sample length, $N$: Dataset examples, $N_{train}$: training examples, $N_{test}$: testing examples, $N_{dev}$: verification examples, $|V|$: vocabulary size, $C$: number of target categories.

| Dataset | $Metrics$ | $L$ | $N$ | $N_{train}$ | $N_{test}$ | $N_{dev}$ | $|V|$ | $C$ |
|---------|-----------|-----|-----|-------------|------------|-----------|-------|-----|
| CR | Accuracy | 105 | 3775 | CV | CV | CV | 5340 | 2 |
| MR | Accuracy | 56 | 10662 | CV | CV | CV | 18765 | 2 |
| MPQA | Accuracy | 36 | 10606 | CV | CV | CV | 6246 | 2 |
| Subj | Accuracy | 120 | 10000 | CV | CV | CV | 21323 | 2 |
| SST-2 | Accuracy | 65 | ~67K | ~65K | 1821 | 872 | CV | 2 |
| SST-5 | Accuracy | 53 | 11855 | 8544 | 2210 | 1101 | 17836 | 5 |

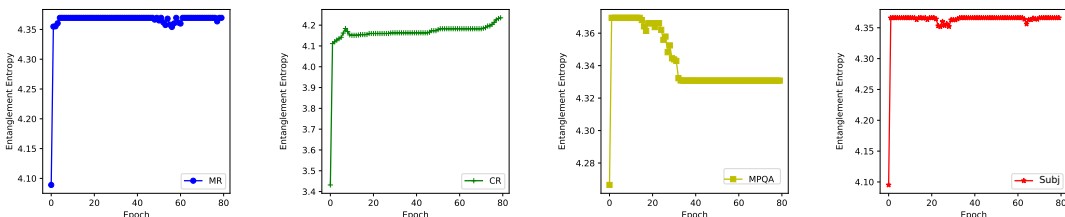

Figure 8: The entanglement entropy in different training epochs on four datasets, including MR, CR, MPQA and Subj when the initialized training.

## D.2 COMPARISON BETWEEN DIFFERENT DIMENSIONS OF PRE-TRAINED WORD VECTORS

As shown in Table 5, we compare the accuracy of TextTN using word2vec (300-dimension) and Bert_large (1024-dimension) pre-trained word vectors, respectively, for nine text classification tasks (e.g., MR, CR and SST-5). The results show that the accuracy for six tasks will have a significantly increase as the dimension of word representations increases in TextTN. Especially for SST-5 and MR, the results of 1024-dimensional word embedding has over 5% increases compared with the results of 300-dimensional word embedding. Comparative experiments show that TextTN has better learning effects for high-dimensional word representations containing more semantic information.

Table 5: Experiments with varying input word embedding dimensions in six classification datasets

| Model | MR | CR | Subj | MPQA | SST-5 | SST-2 |
|-------|-----|-----|------|------|-------|-------|
| TextTN (dim-300) | 82.2 | 85.7 | 95.3 | 90.4 | 48.1 | 91.4 |
| TextTN (dim-1024) | 87.4 | 89.7 | 97.1 | 91.5 | 54.8 | 95.3 |

Table 6: Experiments with different dimensions of probability encoding on Subj dataset.

| Dimension | 2 | 3 | 4 | 5 | 6 | 7 | 8 | 9 | 10 |
|-----------|------|------|------|------|------|------|------|------|------|
| TextTN | 95.3 | 95.0 | 94.5 | 94.2 | 94.6 | 94.4 | 94.7 | 94.6 | 94.4 |

## D.3 A EXPERIMENT ABOUT THE DIMENSION OF PROBABILITY ENCODING

In Section 3.1, we analyze that the dimension $m$ of probability encoding can not exceed 2. In this experiment, we evaluate the conclusion. As shown in Table 6, we compare the accuracy of the TextTN with different probability encoding dimensions set from 2 to 10. In addition, reported results are the average of 10 runs. The results illustrate that when $m = 2$, the accuracy of classification is 95.3, and when $m > 2$, the accuracy has a significant decrease. In particular, the accuracy only achieves 94.2 when $m = 5$, dropping by 1.1%. The Experimental results verify the effectiveness of $m = 2$.

### D.4 THE INITIALIZED TRAINING ON TEXTTN

In order to get the entanglement entropy shown as Eq.10(in this paper) in the initialized training, we set $80$ epochs to training the TextTN on four text classification tasks (i.e., MR, CR, MPQA and Subj). In this process, the entanglement entropy, which is computed according to Definition 1, gradually converges to a determinate value as the increasing of training epoch, which shown as Figure 8.

