# OpenReview forum: "TextTN: Probabilistic Encoding of Language on Tensor Network"
_ICLR.cc/2021/Conference — Reject_

### Official Review · AnonReviewer1 · 2020-10-25

**Rating:** 5
**Confidence:** 2

**Review:**

### Summary
The paper proposes a tensor network for text classification. There are two components: (i) word-GTNs convert word embeddings to m-d probability encoding vectors, and (ii) a sentence-DND takes the word probability encoding vectors as input, combining them using matrix product state (MPS). Experiments on several text classification datasets e.g. SST, CR, MPQA show that the proposed method outperforms existing ones when using word2vec and BERT word embeddings.

### Contributions
The contributions are two-fold:

* The paper proves an effective lower-bound for the bond-dimension. This is then confirmed by experimental results.

* The used sentence-DND combines all possible locations inducing a distribution over the class set. Existing sentence-DNDs employs only location.

###  Pros
I do like theorem 1 about a lower-bound of the bond-dimension based on entanglement entropy. It is very interesting to see how experimental results support the theorem nicely (Figure 3).

The experimental results are a plus.

### Cons

* Clarity:
   - Although the appendices help a lot, the paper isn't easy to read, especially to people who are not familiar with tensor. For instance, what is a "tensor contraction" (right after equation 7)?
   - The paper also doesn't show the complexity. For examples, each d-dim word vector is represented by a \phi(w) tensors, which should have 2^d parameters. In this case, what is the number of parameters of W in equation 7? what is the complexity of the whole model?
   - How to compute entanglement entropy (Appendix D4) isn't provided.

* Originality: In general, the paper doesn't seem to propose significant ideas. The two components are already used.
   - In fact word-GTNs are just a classifier with a fix-sized input. These GTNs are used in computer vision. However, why don't just use a neural network mapping d-dim word embeddings to a distribution over m latent features.
   - The extension of sentence-DTN seems to be trivial (actually I was wondering why no-one hadn't proposed that before). However, the extension should come with more complexity, which wasn't shown in the paper.

* Motivation: The paper isn't convincing why using tensor networks is a good idea. It is unclear how tensors help the task (e.g. what is compositionality here?)

---

> ### Author Response · Authors · 2020-11-22
> **To Reviewer #1**
>
> Thank you very much for your comments and evaluation.
>
> ###Clarity
>
> 1. We add some basic concepts including tensor contraction in Appendix A.1. In a nutshell, the operation of tensor contraction is defined by summing over all of the contracted indices. The contraction of two 2-order tensors is an illustrative example to understand the operation since actually it is same as the operation of matrix multiplication.
>
> 2. The input is represented by a $\phi(w)$ tensors. However, in actual calculations, we do not use $d^N$ parameters. Instead, we initialize low-order tensors (parameters), which are contracted to get the tensor network structure. In Eq.7, assuming that the number of low-order tensors in MPS is $n$, the input dimension of low-order tensors is $m$, and the bond-dimension is $r$, then the size of $W$ is $n \times m \times r \times r$. The specific details are in Appendix A.2, and we have modified Appendix D.4 in the rebuttal version.
>
> 3. In the previous version, we incorrectly pointed the calculation method of entanglement entropy to "Determinate 1". We clarify that the definition and formula of the entanglement entropy is in Definition 1 of the Section 3.2. We have corrected this error in the Appendix D.4 of the rebuttal version.
>
> ###Originality:
>
> For the natural language processing tasks, it is difficult for existing tensor network architectures to select the bond-dimension value automatically.
> Most of existing models need to choose the bond-dimension manually [1][2]. In our  TextTN, we propose to use entanglement entropy to calculate a reference bond-dimension, and then set this bond-dimension in the tensor network. Based on our analysis and experiments, this reference value can result in a good accuracy and efficiency for the text classification task.
>
> Since we can not calculate the entanglement entropy from the data only, we need  GTN and DTN as the prototype of our architecture. We still need to make a few essential improvements on both GTN and DTN in order to identify a reference bond-dimension based on the calculated entanglement entropy. Thus we design word-GTNs to encode the word vectors in a probability distributions, and design sentence-DTN to process the encoded probabilities by word-GTNs, as shown in Section 3.
>
> 1. There are two reasons for why word-GTNs are used instead of other models (e.g., a linear function of the word2vec word embeddings) as the encoders. First, in order to ensure the consistency of the tensor network structure, i.e., the structure of tensor network should be used from the encoder to the classifier, the word-GTNs (a kind of improved tensor network) as encoder are used in TextTN.  Second, word-GTNs as the encoder have some advantages than other some neural network methods. For instance, word-GTNs are  modeled as tensors, which can model complex word dependencies (e.g., the dependence between word dimensions) [3]. In addition, GTN has the properties of high-dimensional separable properties [4] which is helpful for classification tasks.
>
> 2. The original DTN calculates a conditional probability by putting the label bound in a certain position  of the model. The effects of putting the label bound in different positions in MPS are not taken into consideration. Therefore, we propose an all-function approach to model all conditional probabilities, corresponding to all different positions. Experiments also show that this  all-function approach improves the performance that is not trivial. In addition,
> we shall note that our main contribution is that using entanglement entropy to obtain the reference bond-dimension for text tasks in TextTN model.
>
> ###Motivation
>
> For modeling  natural language task, in general, tensor network can capture complex word dependencies in high-order tensor space [3]. Word-GTNs in TextTN can not only capture complex word dependencies, but also has high-dimensional separable properties, which are helpful for text classification tasks [4]. Moreover, according to Theorem 1 and our analysis, we recommend a reference bond-dimension. In experiments, this reference value shows the best performance in Figure 3. This result validates our analysis on the reference bond-dimension. It also shows the advantages of TextTN, in the sense that TextTN can automatically identify optimal hyper-parameters that achieves high  effectiveness on the text classification task. Moreover, compared with different types of neural networks, TextTN  achieves competitive results on six text classification tasks, and achieves SOTA on the SST-5 task.

---

> ### Author Response · Authors · 2020-11-22
> **Reference to  Reviewer #1**
>
> ###Reference
>
> [1] Z. Cheng, B. Li, Y. Fan and Y. Bao, "A Novel Rank Selection Scheme in Tensor Ring Decomposition Based on Reinforcement Learning for Deep Neural Networks," ICASSP 2020 - 2020 IEEE International Conference on Acoustics, Speech and Signal Processing (ICASSP), Barcelona, Spain, 2020, pp. 3292-3296, doi: 10.1109/ICASSP40776.2020.9053292.
>
> [2] Kodryan, Maxim, Dmitry Kropotov, and Dmitry Vetrov. "MARS: Masked Automatic Ranks Selection in Tensor Decompositions." arXiv preprint arXiv:2006.10859 (2020).
>
> [3] Lipeng Zhang, Peng Zhang, Xindian Ma, Shuqin Gu, Zhan Su, Dawei Song. A Generalized Language Model in Tensor Space. AAAI 2019, 7450-7458.
>
> [4] Zheng-Zhi Sun, Cheng Peng, Ding Liu, Shi-Ju Ran, and Gang Su. Generative tensor network classification model for supervised machine learning. Physical Review B, 101(7):075135, 2020.

---

### Official Review · AnonReviewer3 · 2020-10-29
**Timely work, experimental setup has some issues**

**Rating:** 7
**Confidence:** 5

**Review:**

Summary:
A tensor network model for text classification is introduced, which is constructed as the concatenation of a generative matrix product state (MPS) model for low-dimensional word embedding and a discriminative MPS model for classification. This model and different variants are assessed on multiple text classification datasets, with decent performance shown against a range of benchmarks.

Strengths:
The TextTN model seems to be the first tensor network model applied to text classification. In that sense, the experimental results given are important for assessing the usefulness of tensor network models for real-world ML challenges, a question which has seen limited study so far.

The close correspondence shown between the accuracy in classification tasks and the entanglement entropy of the models (Figure 3) is interesting, and hints at the possibility of a compelling link between theoretical quantum many-body physics and practical considerations in ML.

Critiques and Questions:
The word-GTN encoder (which converts high-dimensional word embeddings into low-dimensional inputs to the sentence-DTN) strikes me as being unnecessarily complicated, considering the small role it plays in the model. Given that this just outputs very low-dimensional word embeddings ($d=2$ in the paper), it would make sense to use a trainable word embedding here (or alternately a linear function of the original word embedding) in place of the word-GTN. On that note...

The "TextTN w/o word-GTNs" baseline in the ablation study (Table 3) seems rather misleading, as the paper text describes this as "we directly average the word vectors of words in a sentence to obtain the sentence representation". In other words, not only is the word-GTN removed, but the order of words is lost as well! I would request that this w/o word-GTN baseline to instead use a sentence-DTN whose inputs are low-dimensional vectors given by trainable low-dimensional word embeddings (or alternately a linear function of the word2vec word embeddings). This alternate baseline would make the TextTN significantly simpler (while still remaining a tensor network model), and would also bring it closer to the model of (Stoudenmire & Schwab 2016).

Recommendation:
I would recommend acceptance, owing to the new experimental evidence presented for the performance of tensor networks in NLP. However I do have some doubts about the encoding used for the model architecture, and request that the authors improve their ablation study to better justify the additional complexity coming from two separate linked matrix product state models.

---

> ### Author Response · Authors · 2020-11-22
> **To Reviewer #3**
>
> Thank you very much for your comments.
>
> 1.&nbsp;Word-GTNs have its own properties that a linear function or a trainable word embedding does not have. First,  word-GTNs are modeled as tensors, which can model complex word dependencies (e.g., the dependence between word dimensions) [1]. Second, GTN has the high-dimensional separable properties [2] which is helpful for classification tasks. Third, using word-GTNs can ensure the consistency of the tensor network structure of TextTN.
>
> 2.&nbsp;According to your suggestion, we have improved our ablation experiment in Table3 of the rebuttal paper. We briefly report the results of the experiment here. We now use a linear function to get low-dimensional word representations about "TextTN w/o w-GTNs" in Table 3. The results of accuracy on "TextTN w/o w-GTNs" and "TextTN" are:
>
> | Models | MR| CR | Subj | MPQA |
> |  ----  | ----  |   ----  |     ----  |       ----  |
> |TextTN w/o w-GTNs |  78.6% | 80.2%  | 92.2% | 88.0% |
> | TextTN  | 82.2% | 85.7% | 95.3% |  90.4%   |
>
> Thus, the results show that word-GTNs are better than the linear function.
>
> [1] Lipeng Zhang, Peng Zhang, Xindian Ma, Shuqin Gu, Zhan Su, Dawei Song. A Generalized Language Model in Tensor Space. AAAI 2019, 7450-7458.
>
> [2] Zheng-Zhi Sun, Cheng Peng, Ding Liu, Shi-Ju Ran, and Gang Su. Generative tensor network classification model for supervised machine learning. Physical Review B, 101(7):075135, 2020.

---

### Official Review · AnonReviewer2 · 2020-11-03
**Interesting model; unclear claims; questionable experiments**

**Rating:** 4
**Confidence:** 4

**Review:**

After reading author replies:
I would like to thank the authors to respond to my doubts on some of the results. But I decide to keep the review and the score, because Theorem 1 and Claim 1 are still not well explained. In particular, the explanation like "if the 2nd inequality in Eq. 11 is violated, the network can not capture the amount of information measured by the entanglement entropy " still looks like a conjecture or intuition rather than a mathematical statement.

---------------------------------------------

Originality: High.
The proposed tensor network (TN) based text classification model looks new and interesting.
It consists of two parts: a word-level generative TN model, used to find concise representation of each word; and a sentence-level discriminative TN model, used to classify the sentence based on the outputs of the word-level TNs.
By using the word-level TN for input representation, the dimension explosion problem may be effectively avoided.


Clarity of model description: OK.


Clarity of training method: Low.
The training method of the overall model is not presented. Since there are two TNs, how to train the overall model may not be a trivial problem.


Clarity of analysis: Low.
The major problem of this paper is the analysis presented in Sec 3.2 and Sec 3.3.
I understand that the authors may want to find some theoretical justification on how to choose the bond dimension in the sentence-level TN, as a function of the bond dimension of the word-level TN and the so called entanglement entropy.
However, the result in Theorem 1 is incomprehensible. Only two inequalities are presented, without stating any condition or implication of the inequalities. Since both m and the bond dimension are hyper parameters, what does the 1st inequality mean? Is it a necessary condition on how to choose their values? What happens if the inequality is violated? Even the proof of this theorem does not answer these questions. Same doubts are on the 2nd inequality as well. Additionally, how does one even know what the entangle entropy of a model is before training the model?

The statement of Claim 1 is even more problematic. Not able to understand what it says.


Clarity of expeiremtal results: OK, but not clear enough.
The authors claim that when combined with BERT for word embedding, the proposed model can outperform the SOTA methods.
However, there are several things that are not clear.
1. It would be more fair to compared the combined model with BERT with some similar models that also use BERT for word embedding.
2. It is claimed that the proposed method is better than word-GTN. But word-GTN is a unsupervised learning model. Why is it meaningful to make such a comparison?
3. In the introduction, another TN based method "TSLM" is mentioned. Would it be more fair to compare the proposed method with TSLM, as both of them use TN for modeling?


Overall, the proposed model looks interesting and shows some potential improvements over SOTA. However, the quality of the paper is degraded by the unclear claims and some questionable experimental results.

---

> ### Author Response · Authors · 2020-11-22
> **To Reviewer #2**
>
> Thanks a lot for your comments and advice.
>
> ###Clarity of of training method:
>
> We add an Algorithm 1 to describe the training process of TextTN,  in the end of Section 3.3 of the rebuttal version.
>
> ###Clarity of  analysis:
>
> About Theorem 1, we make further explanations on it. In TextTN, suppose the entanglement entropy, which measures the communication information [1],  equals to $E$. The bond-dimension of TextTN should satisfy Eq. 11, as proved in Appendix B.  As for the violation of the 1st inequality, if the bond-dimension is larger than $m^{n/2}$,  then TextTN will be rank deficient [2], which do not meet the constraints shown as the Appendix A.2 and Appendix B in the rebuttal version. Consequently, the needed bond-dimension should be no larger than $m^{n/2}$. On the other hand, if the 2nd inequality in  Eq. 11 is violated, the network can not capture the amount of information measured by the entanglement entropy $E$.
>
> In our paper, we choose the lower bound $2^E$ as the reference bond-dimension, considering it is the smallest value satisfying Eq.11. Comparing with $2^E$, choosing a larger value which  satisfies Eq. 11 will increase the number of parameters and the whole computational cost of the TextTN, but only gain little information from the data. The increase of the bond-dimension will increase the model complexity, possibly leading to the overfitting problem.  We have added detailed description after the Theorem 1 in the rebuttal version.
>
> As for calculating the required entanglement entropy $E$, we first need to use a sufficiently large bond-dimension (e.g., 100). Since the initial bond-dimension is large enough and the entanglement entropy finally converges, the $E$ measures the amount of information that the network can capture from a certain data set [1].
> After calculating the entanglement entropy $E$, according to the proof of the Theorem 1, we get the lower bound of bond-dimension shown as Eq.11, which is the reference bond-dimension. We clarify this process in the penultimate paragraph
> of Section 3.2 in the updated paper.
>
> Moreover, in the experiments of Appendix D.4, take the MPQA data set as an example, the result shows that the entanglement entropy will eventually converge to a value $E=4.32$. In the experiment of Figure 3, the result shows that when $2^{E}=19$ as the reference bond-dimension, TextTN obtains the best accuracy of MPQA on TextTN. The experiment results also verifies our analysis.
>
> ###The statement  of Claim 1
>
> Using the Claim 1, we want to stress the limitation of original DTN and many existing tensor networks. In existing methods, the effects for the different  positions of label bound in MPS can not taken into consideration. Specifically, in the original DTN, a conditional probability is only calculated by putting the label bound in a certain position of the DTN. Therefore, we propose an all-function approach to model all conditional probabilities, corresponding to all different positions. Experiments also show that this all-function approach improves the stability of model predictions of TextTN.
>
> ###Clarity of expeiremtal results
>
> 1. In Table 1, since all the baselines does not use BERT embeddings, for a fair comparison,  we also do not use BERT, but the word2vec pre-trained embedding as used in the reported baselines. In Table 2, since the baselines (e.g., BERT, CNN\_Large) use BERT embedding, we also evaluate TextTN with BERT.
>
> 2. Word-GTNs and the existing GTN are two different models.
> Word-GTNs are  proposed in our paper, and can be considered as the modified version of GTN.
> In TextTN, word-GTNs,  as the probability encoders, are used to encode words. GTN is first proposed as an unsupervised learning model [1]. However,it can also be used for supervised learning by changing the output method [3].
>
> 3. TSLM is a theoretical tensor network language sequence model, but due to exponential parameter problems, it does not actually use tensor network, which we have already mentioned in the introduction. What is more, TSLM is  evaluated by language modeling tasks, and TextTN is a model for text classification, so that the comparison between the TSLM and TextTN could  not be straightforward.
>
> [1] Yoav Levine, Or Sharir, Nadav Cohen, and Amnon Shashua. Quantum entanglement in deep learning architectures. Physical review letters, 122(6):065301, 2019.
>
> [2] Ivan Oseledets.  Tensor-train decomposition.SIAM Journal on Scientific Computing, 33(5):2295–2317, 2011.
>
> [3] Zheng-Zhi Sun, Cheng Peng, Ding Liu, Shi-Ju Ran, and Gang Su. Generative tensor network classification model for supervised machine learning. Physical Review B, 101(7):075135,2020.

---

### Official Review · AnonReviewer5 · 2020-11-09
**Official Blind Review #5**

**Rating:** 6
**Confidence:** 2

**Review:**

The paper proposed TextTN that applied Tensor Network on text input. They effectively solved the high dimensionality problem and achieved the state of the art on multiple text classification problems. The paper also proved the bond dimension and showed experiment results to justify their theory.

In the experiments in table 1, was there a reason why you don't use the pretrained BERT embeddings? Can you compare with the BERT baseline as well?

In your BERT + TextTN experiment in table 2, do you finetune the pretrained BERT embeddings?

Could you please analysis on the efficiency of your model? It could be a benefit of TextTN over the baseline models.

The results on some experiments over the baseline models are not very significant (by ~0.1). I am concerned how much of them comes from hyper-parameter tuning vs. model structure. It's possible that the tasks are not very hard so there are not too much headroom. Can you evaluate it on harder tasks such as other tasks in the GLUE benchmark?

---

> ### Author Response · Authors · 2020-11-22
> **To Reviewer #5**
>
> Thank you very much for your comments and suggestions.
>
> 1.&nbsp;In Table 1, for all the baselines in comparison, they use Word2Vec embeddings, rather than pre-trained BERT embeddings. For a fair comparison, TextTN also used the Word2Vec embeddings as the input word vectors. In Table 2, we carry out BERT+TextTN experiments using pre-trained BERT embeddings, since the BERT and other pre-trained model (e.g., ELMo) are selected as the baseline models.
>
> 2.&nbsp;In BERT+TextTN, we did not finetune the pre-trained BERT embeddings.
>
> 3.&nbsp;The comparative results between TextTN and baselines for accuracy Acc (%) ,epoch time T (s/epoch) and parameter θ on SUBJ dataset are shown as follows:
>
> | Models | $Acc$ | $T(s)$ | $\theta$ |
> |  ----  | ----  |   ----  |     ----  |
> |CNN |  93.4 | 3.2  | 1.4M |
> | DiSAN   | 94.2 | 6.7 | 2.3M |
> | MPSAN  | 94.6  | 5.6 | 1.9M |
> | TextTN  |  95.3 | 4.1  | 0.56M  |
>
> In the table, the parameter amount of TextTN (0.56M) is smaller than that of CNN (1.4M), DiSAN (2.3M) and other baselines, and the accuracy of TextTN is higher than baselines. As for time efficiency, during the training process, each epoch has 80 samples. On TextTN, the running time of each epoch is 4.1s, which is slightly higher than the running time of CNN (3.2s). However, the training speed of TextTN is 38.8% faster than DiSAN and 26.8% faster than MPSAN.
>
> 4.&nbsp;We evaluated TextTN on the SST-2 dataset belonging to the GLUE benchmark, and obtained a good result which is shown in Table 2 of the submitted paper. In addition, since the baselines on GLUE leaderboard are usually based on large-scale pre-trained models such as BERT, we are also testing other tasks in the Glue benchmark on "BERT+TextTN" model. Due to time constraints of rebuttal version, to be honest, we have not been able to find suitable hyperparameters (e.g., dropout, learning rate and batch size) to make the "BERT+TextTN" model achieve satisfying results.
>
> With respect to the hyperparameters of TextTN, we would like to clarify that our focus in this paper is to analyze the bond-dimension of TextTN. Based on Theorem 1 and its analysis, we can calculate a reference bond-dimension, which is obtained based on the entanglement entropy. The results of Table 1 and Figure 3 in our paper show that, among all the tested bond-dimension settings, this reference bond-dimension calculated automatically (rather than selecting manually), achieves the best effects of TextTN on different datasets.
>
> As for "BERT+TextTN" on GLUE benchmark tasks, it will involve many other hyperparameters that can affect the effectiveness of text classification. Therefore, we will investigate the model architecture and hyperparameter settings of BERT+TextTN in depth and carry out a series of experiments in future work.

---

### Decision · Program_Chairs · 2021-01-07
**Final Decision**

**Decision:**

Reject

**Comment:**

While the submission has promising components, the reviewers were not able to reach a consensus to recommend acceptance. The main concerns is that (1) theorem statements and assumptions are not clearly explained, and (2) the novelty of the approach is not made clear, and (3) there remain concerns on whether the experimental results are due to hyperparameter search or improvements due to the model.